# Physical Aging Behavior of the Side Chain of a Conjugated Polymer PBTTT

**DOI:** 10.3390/polym15040794

**Published:** 2023-02-04

**Authors:** Tengfei Qu, Fanzhang Meng, Linling Li, Chen Zhang, Xiaoliang Wang, Wei Chen, Gi Xue, Evgeny Zhuravlev, Shaochuan Luo, Dongshan Zhou

**Affiliations:** 1State Key Laboratory of Coordination Chemistry, Department of Polymer Science and Engineering, School of Chemistry and Chemical Engineering, Nanjing University, Nanjing 210023, China; 2University and College Key Lab of Natural Product Chemistry and Application in Xinjiang, School of Chemistry and Environmental Science, Yili Normal University, Yining 835000, China; 3Institute of Critical Materials for Integrated Circuits, Shenzhen Polytechnic, Shenzhen 518055, China; 4School of Chemistry and Materials Science, Nanjing University of Information Science & Technology, Nanjing 210044, China; 5School of Environment and Energy, Peking University Shenzhen Graduate School, Shenzhen 518055, China

**Keywords:** calorimetry, physical aging, side chain, PBTTT

## Abstract

This paper provides a viewpoint of the technology of the fast-scanning calorimetry with the relaxation behavior of disordered side chains of poly[2,5-bis(3-dodecylthiophen-2-yl)thieno[3,2-b]thiophene] (PBTTT-C12) around the glass transition temperature of the side chains (*T*_g_*_,γ_*). PBTTT is an ideal model of the high-performance copolymer of poly(alkylthiophenes) with side chains. The *γ*_1_ relaxation process of the disordered side chains of PBTTT was detected as a small endothermic peak that emerges before the *γ*_2_ relaxation process. It shows an increase with increasing temperature as it approaches the glass transition temperature of the disordered side chains of PBTTT. The ductile–brittle transition of PBTTT in low temperatures originating from the thermal relaxation process is probed and illustrated by physical aging experiments. The signature is shown that the relaxation process of the disordered side chain of PBTTT at low temperatures varies from Arrhenius temperature dependence to super Arrhenius temperature dependence at high temperatures. These observations could have significant consequences for the stability of devices based on conjugated polymers, especially those utilized for stretchable or flexible applications, or those demanding mechanical robustness during tensile fabrication or use in a low-temperature environment.

## 1. Introduction

Conjugated polymers have useful electronic [1], photonic [2], and thermoelectric [3,4,5] properties which are used in various flexible devices. Conjugated polymers are often composed of side chains and main chains, such as poly(3-alkylthiophene) [6,7], poly[2,5-bis(3-dodecylthiophen-2-yl)thieno[3,2-b]thiophene] (PBTTT) [8,9], and diketopyrrolopyrrole-based polymers [10,11,12,13]. The side chain improved the processing technology [14] and ameliorated the glass transition temperature [15] of conjugated polymers. It has become a research hotspot to control the morphology and properties of conjugated polymers through side-chain engineering [16]. The different aggregation states of the side chain of the conjugated polymer PBTTT mainly affect the microstructure in two aspects. On the one hand, it was found that the side chains would form an interdigital ordered aggregate, and the size of the interdigital structures was related to the side chain tilt angle [17], and the processing [18]. Compared with the side chains of PBTTT without interdigitation, the interdigitated state of the part of the side chains is the preferred state at room temperature [17], so that local nanocrystalline domains could be formed eventually. On the other hand, there is a disordered fraction of side chains in the aggregation state of conjugated polymers, and the lamellar spacing with non-interdigitation side chains becomes more prominent than that with interdigitation side chains from X-ray diffraction or atomic force microscopy measurements [17,19]. The lamellar spacing was increased by solution shearing due to increased disorder of side chains. Furthermore, their mechanical property under variable morphologies of side chains also cannot be ignored for it is closely associated with the device’s performance [20,21] and the polymer’s chemical structure [22,23]. Increasing the temperature of the rubbing process generates a more disordered fraction of alkyl side chains of PBTTT. Additionally, the plasticity of the polymer in mechanical properties is increased because the temperature has an important influence on the size of the oriented domains following the main chain direction and alongside the side chains [20]. For the different chemical units of regioregular poly(3-hexylthiophene) and poly[N-9′-heptadecanyl-2,7-carbazole-alt-5,5-(4′,7′-di-2-thienyl-2′,1′,3′-benzothiadiazole)], the frozen state and relaxation of side chains of regioregular poly(3-hexylthiophene) are crucial contributions to the toughness change in channel-like cracks at 243 K and 163 K. Similarly, the fracture behaviors and the crack features of poly[N-9′-heptadecanyl-2,7-carbazole-alt-5,5-(4′,7′-di-2-thienyl-2′,1′,3′-benzothiadiazole)] at room temperature and 243 K were attributed to the relaxation of the branched side chains [22].

In recent years, researchers proved the close relationship between the mechanical relaxation process and the side chains of conjugated polymers [24,25,26]. In the dynamic mechanical analysis studies of conjugated polymers [16], the side chain transition temperature has an important influence not only on the side chain motion but also on the mechanical parameters [27] and other stretchable devices’ properties [28,29]. It was found that a greater drop in modulus change after side-chain relaxation due to the side-chain density increased, and the more mobile structure of side chains induces a drop in several conjugated polymers [30]. Polymers are considered to show moderate ductility at temperatures below the glass transition temperature of the disordered side chains (*T_g,γ_*). Below their secondary glass transition temperatures, the deformation of the conjugated polymers occurs through active slip systems with dissipating energy [31]. In addition, the side chains of conjugated polymers play a dominant role in the extent of mechanical relaxation [30]. The research indicated that the relaxation temperature of the side chains (241 K) of non-thiophene-linked fluorinated benzotriazole is lower than the one (249 K) of poly{4-(5-(4,8-bis(3-butylnonyl)-6-methylbenzo[1,2-b:4,5-b′]dithiophen-2-yl)thiophen-2-yl)-2-(2-butyloctyl)-5,6-difluoro-7-(5-methylthiophen-2-yl)-2H-benzo[d][1,2,3]triazole} (PBnDT-FTAZ) due to the higher density of the side chains in a given volume. The thiophene rings of side chains in PBnDT-FTAZ provide a less free volume [30].

However, the study on the motion behavior and the thermal relaxation characteristics of disordered side chains of conjugated polymers below *T_g,γ_* is still unclear. The disordered side chain structure is in a metastable state, and the time scale of side chain segmental motion is often very short, requiring a fast-scanning device to capture. The formation of a disordered state of side chains requires a high cooling rate from the molten state. Therefore, studying the relaxation behavior of disordered side chains below *T_g,γ_* is challenging work. In this study, fast scanning calorimetry (FSC) was used to control the side chain in different disordered states. PBTTT, unlike the P3HT, has dense side chains, a model-conjugated polymer that consists of an alkyl side chain and a thiophene main chain. The FSC has the characteristics of extremely high heating and cooling rates [32,33], which can not only probe the crystallization process of different arrangements of polymer chains around the temperature of glass transition by isothermal crystallization but also is capable of the requirement of rapidly quenching the side chains of PBTTT in the molten state to the disordered state. Without the thermal degradation to the sample, it is available to detect the relaxation behavior of the side chains of PBTTT in a broad time and temperature scale. Our research provides systematic insight into conjugated polymer structure and related thermal relaxations around the *T_g__,γ_*.

In this study, we identified the motion behavior of the side chain segment from lower temperatures to higher temperatures. It was found that the disordered side chain is frozen at low temperatures. With the temperature increase from below 233 K, the motion behavior of the side chain segment varies from Arrhenius behavior (*γ*_1_) to super Arrhenius behavior (*γ*_2_). This transition temperature helps us to divide the two relaxation behaviors (*γ_1_, γ_2_*) of the disordered side chains of PBTTT. Namely, the local motion behavior of the side chain segment and the motion behavior of the whole side chain. Further, such two kinds of relaxation behavior help us obtain the glass transition temperature of the disordered side chain. The study of the relaxation behavior of the disordered side chain is beneficial for the understanding of the ductile–brittle transition process of side chains of PBTTT.

## 2. Materials and Methods

### 2.1. Materials

PBTTT (Mw = 31,800, Ð = 1.9) conjugated polymer was supplied by Luminescence Technology Corp (New Taipei City, Taiwan).

### 2.2. Methods

Different temperature annealing studies of the sample were performed using the custom-made ultrafast FSC device in combination with XI-395 sensors (Xensor Integration, EJ Delfgauw, The Netherlands) calibrated by indium. The XI-395 sensor has a thermally active area of 60 μm × 70 μm. The sample of PBTTT was cut under the optical microscope by a purified scalpel and then put on the center of the temperature control area of the sensor by a thin copper wire. Because of the tiny sizes of the thermally active area and the sample, controlled heating (cooling) rate of up to 10,000 K/s can be achieved using liquid nitrogen as a refrigerant. All measurements were performed under an atmospheric nitrogen atmosphere. Without otherwise specified, the cooling and heating rates of FSC used in this paper were 10,000 K/s and 10,000 K/s, respectively.

The temperature–time profile for studying the physical aging behavior of PBTTT is presented in Figure 1a. Due to the tiny sampling amount on the chip sensor (60 μm × 70 μm, Figure 1b), a well-controlled scanning rate of up to 10,000 K/s can be achieved by FSC.

## 3. Results

Heat capacity curves at the indicated aging times and the following aging temperatures are shown in Figure 2. In all cases, the area of heat capacity peaks increases with aging time induced by the segmental motion of the side chain. Aging at temperatures from 173 K to the *T_g,γ_*, the heat capacity peak progressively shifts to higher temperatures. At 223 K, for short aging times, the heat capacity peaks are well separated from the step at the *T_g,γ_*. However, it progressively overlaps with such steps for longer aging periods, similar to aging at higher temperatures. When aging at 183 K, the heat capacity peaks show significant deviation from the step at the glass transition range, at an even lower temperature range.

With the increase in temperature, the flexibility of the linear polymer segment increases, and the mobility of the segment increases [34]. A conjugated polymer of PBTTT can be regarded as a series of subsystems formed by the dodecyl side chains. Each side chain has its statistical behavior and establishes an equilibrium with the main chains or other side chains. The transformation is divided into two categories based on the simplification of this empirical view [35].

The metastability of polymer glasses is proven by relaxation processes which straightforwardly may be analyzed by annealing experiments [36]. In one of the approaches [36], the time evolution of the decrease in the enthalpy (*H*) during annealing can be described using Equation (1)
(1)$$∆H_{relax}\left(T_{a},t\right)=∆H_{relax,max}\left(T_{a}\right)[1-\exp[-t/\tau {]}^{\beta }]$$

Here, *t* and *τ* are the annealing time and characteristic relaxation time constant of the motion structure units of the side chains, respectively, and β is the Kohlrausch–Williams–Watts parameter [37,38,39,40,41].

Generally, in all the previously described thermal procedures, the amount of recovered enthalpy of glass for a profile of time *t_a_* at a given temperature *T_a_* was evaluated by integration of the difference between thermograms of aged and unaged samples, according to the relation [42,43]
(2)$$∆H_{relax}\left(T_{a},t_{a}\right)=\overset{T_{y}}{\underset{T_{x}}{\int }}(C_{p}^{a}(T)-C_{p}^{u}\left(T\right))dT$$
where $C_{p}^{a}\left(T\right)$ and $C_{p}^{u}\left(T\right)$ are, respectively, the specific heat of the aged and unaged samples and *T_x_* and *T_y_* are, respectively, temperatures (appropriately chosen) below and above the calorimetric *T_g_*. Δ*H_relax,max_*, the difference between the enthalpy of the glass at the beginning of the annealing experiment and that of the extrapolated liquid state at the annealing temperature, that is, the maximum possible enthalpy of relaxation. The latter value depends on the difference between the heat capacities of the vitreous and liquid states at the temperature of vitrification, Δ*C_p_*, and the difference between *T_g_* and the annealing temperature *T_a_* [44].

The *γ*_1_ relaxation process (the small green ellipse represents the local motion of the partial segment of the disordered side chain, and the light green area is the *γ*_1_ relaxation region) of disordered side chains of PBTTT follows the Arrhenius temperature dependence (green line), and the *γ*_2_ relaxation process (the local motion of the partial segments of the disordered side chains are shown in little green ellipses, and the light blue area is the *γ*_2_ relaxation region) of disordered side chains follows the supper Arrhenius temperature dependence (red line). The dark blue curves represent the main chains of PBTTT as shown in Figure 3.

An overview of the kinetics of equilibrium recovery of disordered side chains of PBTTT, in terms of the time scales to reach equilibrium calculated by the KWW equation, *τ_eq_*, is shown in Figure 3. At lower temperatures, i.e., far from the glass transition temperature of the disordered side chains of PBTTT, only on a time scale associated with the monotonous decay to equilibrium, an Arrhenius behavior can be observed associated with the *γ*_1_ relaxation of disordered side chains of PBTTT. At higher temperatures, with *τ_eq_* decreasing in a super Arrhenius behavior that related to the *γ*_2_ relaxation of disordered side chains, and a rapid time scale exhibiting temperature dependence and seemingly decreasing enthalpy changes with increasing temperature is displayed in Figure 4. This result suggested that the cooperativity rearranging region varies from the partial motion behavior of the side chain segment to the motion behavior of the whole side chain [45]. So, it demonstrated experimentally that the glass transition temperature of the disordered side chain is 233 K. The side chain of PBTTT in the disordered state has a small amount of free volume that permits only the local motion of the partial segment of the side chain (i.e., the *γ*_1_ relaxation process is shown in Figure 3) below the *T_g,__γ_*. The *γ*_1_ relaxation of the disordered side chains of PBTTT is a thermally activated process and can be initiated above certain temperatures. An Arrhenius process is shown as a straight line in Figure 3. This process can be regarded as generated from the non-cooperative rearrangement of local segments of the disordered side chains in large regions, as shown by the purple arrows in the small green ellipses in Figure 3. In the regions, a rigorous requirement of the cooperative motion has a relatively fixed orientation of a large amount of the disordered side chains, such as the glassy state by rapidly quenching from the molten state. So, the partial segments of the disordered side chains in local motion are located as “islands” of the “islands mobility” in the *γ*_1_ relaxation.

As the temperature increases and exceeds the *T_g,__γ_*, the segmental motion of disordered side chains becomes more intense, the length of the mobile segment increases, and the free volume rises, initiating the *γ*_2_ relaxation process (the process is shown in Figure 3) of the disordered side chains. This process needs a great degree of cooperativity between the segments of disordered side chains and side chains. As the free volume grows, temperature increases, allowing the side chains to relax more independently. Here, the *γ*_2_ relaxation structure lives the *γ*_1_ relaxation process. It is a temporary distribution of locally *γ*_2_ relaxation optimized real-space arrangements after the cooperativity motion. The *γ*_1_ relaxation process decays at the *γ*_2_ relaxation process due to the *γ*_1_ relaxation process correlation in space and time. The motion of partial segments of the disordered side chains in the *γ*_2_ relaxation process is related to the cooperative motion, as shown by the black arrows among the light blue ellipses in Figure 3. The *γ*_2_ relaxation process depends on the cooperativity of the high mobility of adjoining “islands”. Double time (*t* and *τ*) correlations describe the time-dependent dynamics of side chain motion structure units under different annealing temperatures in Equation (1). The enthalpy changes (∆H) in the corresponding segmental motion of the side chain at different annealing temperatures are displayed in Figure 4. In all cases, the enthalpy changes increase with aging time induced by the segmental motion of the disordered side chain. Aging at temperatures from 173 K to 233 K results in the enthalpy changes progressively shifting to higher values. Aging at temperatures higher than 233 K, the enthalpy changes gradually shift to lower values. When the conjugated polymer PBTTT is cooled through 233 K, the segments of the disordered side chains lose their flexibility. However, above 233 K the free volume has shrunk to a low value that the motion of the flexible segment of the side chains is slightly hindered. In addition, the slopes of ∆H values verse aging temperature show that the speed of the *γ*_1_ relaxation of the side chain segmental motion is slower than the one of *γ*_2_ relaxation. The mobility of the disordered side chain is enhanced at high temperatures. That suggested the size of cooperativity rearranging regions from the partial segment motion behavior of the side chain extend to the motion behavior of the whole side chain.

The parameter *β* in Equation (1) may be regarded as reflecting the breadth of distribution of relaxation times or as a direct measure of the departure from an exponential decay function [46]. In this case, changes in *β* with temperature correspond to changes in the degree of cooperativity of disordered side chains of PBTTT. Here, we interpret *β* as a measure of cooperativity, which can be usefully defined in terms of the number of side chain segments involved in a particular relaxation process. Low values of *β* correspond to a high degree of cooperativity and a large number of side chain segments. If it is assumed that the activation enthalpy per segment is a weak function of the side chain, the involvement of an amount of side chain segments would result in a large activation enthalpy for the relaxation process. The observed inverse correlation between *β* (Figure 5), enthalpy changes (Figure 4), and aging temperature are consistent with the assumption, indicating that the activation energy per segment is indeed a weak function of side chains of PBTTT. Some support for this reasoning is found in the theory proposed by Bendler and Ngai [47]. This theory provides theoretical support for the KWW function, with *β* determined by the number of correlated low energy states in the heat bath around a reference molecule and by the strength of coupling between the heat bath and conformational states of the side chains. The relationship between *β* and the observed temperature is also consistent with the glass transition temperature of the disordered side chain, as shown in Figure 3.

An approach to obtain insights into the physical aging behavior relies on the structure recovery process of glassy state polymer. The fictive temperature, *T_f_*, describes the structure state of glassy material that is implied the effects of thermal treatments. The thermodynamic state in terms of *T_f_* is shown in Figure 6 as a function of the aging temperature at the indicated time. Close to *T_g,γ_*, aging exhibits the standard behavior consisting of a monotonous decay toward equilibrium, which is marked by the condition *T_f_* > *T_ag_*. Far more, at higher temperatures, specifically 223 K in Figure 6, aging takes place with a rapid decay rate to reach the plateau with *T_f_* = *T_ag_*.

## 4. Discussion

Physical aging is an important phenomenon in polymers, inorganic glasses, and some metals. The aging of the conjugated polymer has to be considered in the testing of plastics, especially in the prediction of their mechanical behavior in the long term. The very property of the side chain of conjugated polymer PBTTT that changes during aging is segment mobility. The structure of segment motion and the relaxation times are directly associated with segment mobility. According to the above, when the conjugated polymer sample is heated to above *T_g_*, especially in a molten state, it readily reaches thermodynamic equilibrium. The history of the disordered side chains of PBTTT has been “forgotten”, any previous relaxation process of the disordered side chains by aging experiments that occurred below *T_g_* has been erased. Therefore, the aging of the disordered side chains is a thermo-reversible process that can be repeated a large number of times on the same PBTTT sample for the research of the relaxation process of the disordered side chains.

The experimental phenomenon of physical aging of the conjugated polymer PBTTT below *T_g,γ_* mentioned in the above triggered by the relaxation of the side chain segment, in which the typical time scale decreases to feasible values not too far below *T_g,γ_*. The importance of side chain relaxation on device stretchability of the polymer films cannot be ignored for the thermal relaxation process of chain segments [45]. In all, the thermomechanical behavior under low applied strain is well related to the stress–strain behavior of the films. Additionally, the aging temperature substantially impacts the aging process and the mechanical behavior of polymer films [48]. A retraction of this embrittlement was observed when the aging temperature was further increased below *T_g_*, indicating that the aging embrittlement is possibly related to the segmental relaxation of the side chain.

In addition, a physical aging approach is carried out to analyze the brittle–ductile transition and mechanical properties of polylactic acid (PLA). Due to the reversible transformation of the disordered structure of PLA at low temperatures, different stable aggregate states are formed by physical aging [49]. Therefore, controlled physical aging was used as a new approach to modify the brittle–ductile transition of materials. Through observed changes in relaxation behavior, the thermal relaxation process in PBTTT with side chains by physical aging was reported in this work. The relaxation of the side chains of PBTTT at low temperatures has been probed. Due to the non-equilibrium nature of the glassy side chain of PBTTT, the thermodynamic parameters and physical properties vary continuously with time, even when annealing below the *T_g_* caused by reversible rearrangement of disordered side chains. For ductility over a wide temperature range, the relaxation of the side chains of the conjugated polymer has to lie at temperatures as low as possible. So, one way to enhance the ductility is broadening the temperature range of the brittle–ductile transition of side chains in which the conjugated polymer is sensitive to aging and thermal history.

## 5. Conclusions

The mechanical relaxation experiments of a conjugated polymer flexible device are associated with the conjugated polymer chemical structure and structure recovery over a wide range of temperatures. The ductile–brittle transition in these polymers is strongly affected by temperature. The ductile–brittle transition of PBTTT in low temperatures originated from the thermal relaxation process and is probed and illustrated by physical aging experiments. The *γ*-relaxation of the side chain of PBTTT at low temperatures is shown the side chain segmental motion from Arrhenius temperature dependence to the super Arrhenius at higher temperature dependence. The importance of low-temperature mobility of the side chain of PBTTT has been emphasized, and it has been systematically studied connected with detecting the relaxation of the disordered side chains and measuring their behaviors of the different segmental motions. All in all, we believe that the relaxation of the disordered side chain of PBTTT should be paid more attention to when studying the physical aging of conjugated polymers. On the one hand, it does not seem likely that the interdigitation of side chains would demonstrate any measurable melting point of nanocrystal domains. On the other hand, it also exhibited a different type of segmental motion in the sizes of cooperatively rearranging regions that were associated with the mechanical relaxation experiments of conjugated polymer flexible devices in low temperatures.

## Figures and Tables

**Figure 1 polymers-15-00794-f001:**
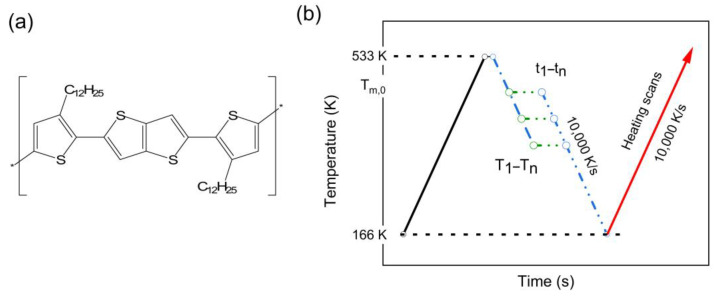
Different temperature annealing information of PBTTT conjugated polymer. (**a**) Chemical structures of PBTTT. (**b**) Temperature–time profile for generation and analysis of annealing peaks in FSC heating scans.

**Figure 2 polymers-15-00794-f002:**
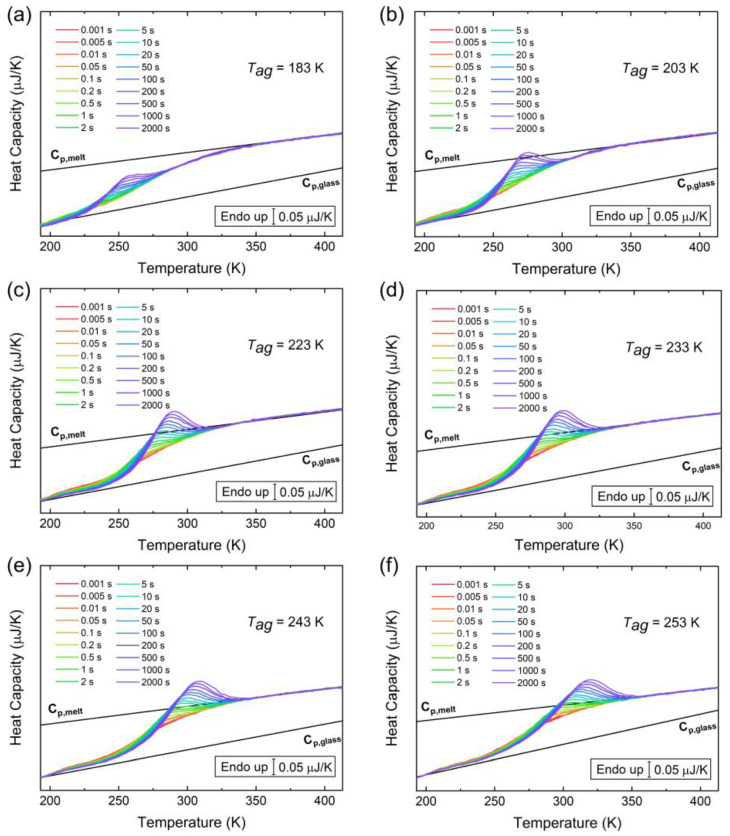
Heat capacity curves at the indicated aging times and the following aging temperatures: (**a**) 183 K; (**b**) 203 K; (**c**) 223 K; (**d**) 233 K; (**e**) 243 K; and (**f**) 253 K.

**Figure 3 polymers-15-00794-f003:**
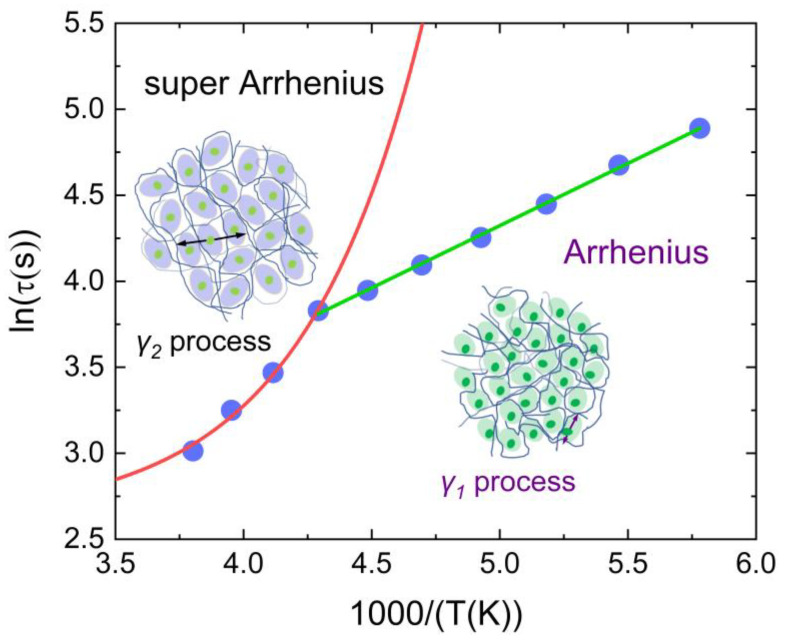
Schematic representation of temperature dependence for relaxation times. The small green ellipse on the right represents the local motion of the partial segment of the disordered side chain; the light green area is the *γ*_1_ relaxation region; the purple arrows represent the non-cooperative rearrangement of the local segment of the disordered side chain; the green line represents the *γ*_1_ relaxation disordered of the side chains of PBTTT following the Arrhenius temperature dependence. The little green ellipses on the left represent the local motion of the partial segments of the disordered side chains; the light blue area represents the *γ*_2_ relaxation region; the black arrows among the light blue ellipses represent the cooperative motion of partial segments of the disordered side chains; the red line represents the *γ*_2_ relaxation of the disordered side chains follows the supper Arrhenius temperature dependence. The dark blue curves represent the main chains of PBTTT.

**Figure 4 polymers-15-00794-f004:**
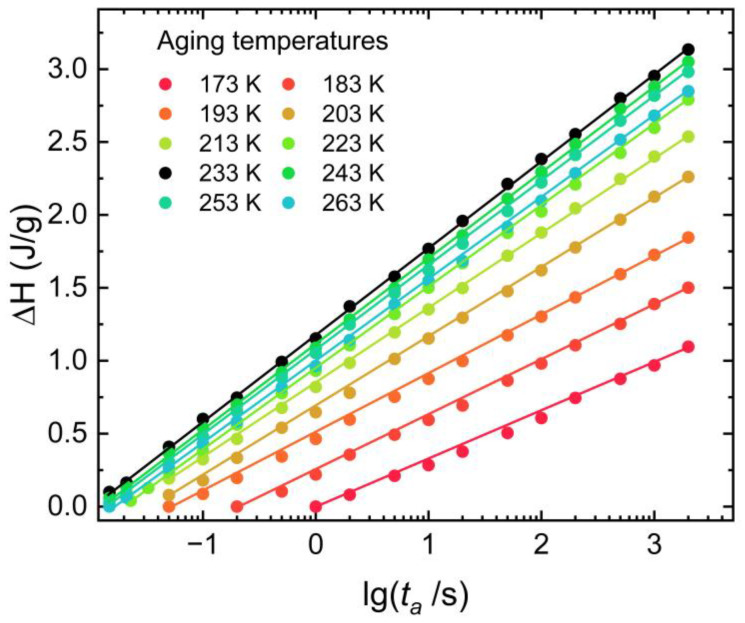
∆H values at different aging temperatures of PBTTT.

**Figure 5 polymers-15-00794-f005:**
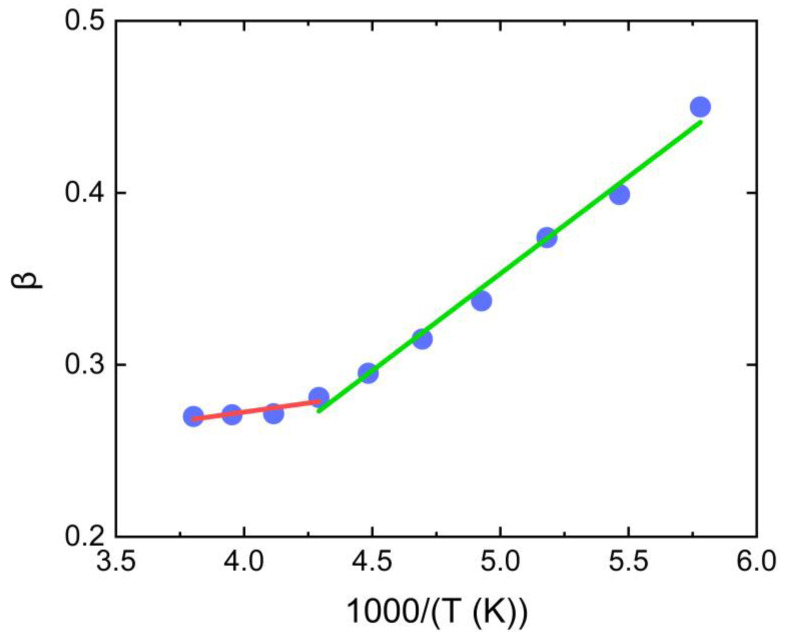
Schematic representation of the correlation between temperature and *β*. The *β* of *γ*_1_ relaxation of disordered side chains follows lower temperature dependence (green line), and the *β* of *γ*_2_ relaxation of disordered side chains follows higher temperature (red line).

**Figure 6 polymers-15-00794-f006:**
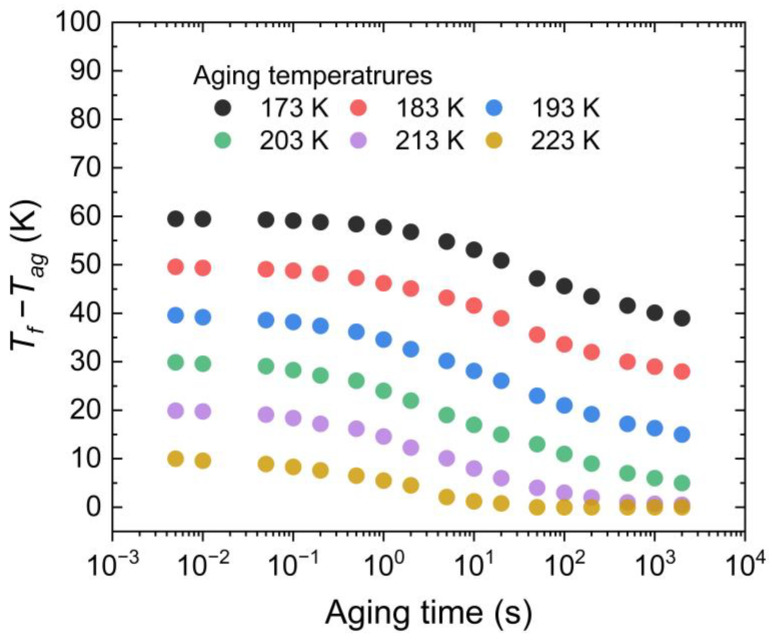
Evolution of *T_f_* taken as the distance from *T_g,γ_* with the aging time for PBTTT aged at indicated adding temperatures.

## Data Availability

Not applicable because of further research.

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
