# Peer review of "Physical Aging Behavior of the Side Chain of a Conjugated Polymer PBTTT"

_polymers, 2023, doi:10.3390/polym15040794_

Round 1
Reviewer 1 Report
This work, "Physical Aging Behavior of the Side Chain of a Conjugated Polymer PBTTT," discusses fast-scanning calorimetry and the relaxation behavior of disordered side chains of poly[2,5-bis(3-dodecylthiophen-2-yl)thieno[3,2-b]thiophene] (PBTTT-C12) below their sub-Tg. The optimal high-performance copolymer of poly(alkylthiophenes) with side chains is PBTTT. Physical aging experiments demonstrate the thermal relaxation-induced ductile-brittle transition of PBTTT at low temperatures. Furthermore, the disordered side chain of PBTTT's -relaxation process at low temperatures is Arrhenius temperature-dependent and super-Arrhenius at high temperatures. These findings may affect the stability of conjugated polymer devices, particularly those used for stretchable or flexible applications, tensile construction, or low-temperature use. Therefore, I would recommend it be published in Polymers after the following minor issues are addressed:
- The abbreviation should be treated more carefully. The authors should check again to ensure that they define the abbreviation when mentioned first in the article. Then use the abbreviation in the following part. In addition, the authors could refrain from using abbreviations if they mention them a few times in the article.
- The font, italics, superscripts, and subscripts should be consistent in the whole manuscript.
- In figure 1a, the resolution of the chemical structure should be revised.
- For the introduction, several works of DPP-type polymers and packing properties of thiophene-based polymers should be cited — [ J. Am. Chem. Soc. 2011, 133, 7, 2198–2204; Organic Electronics 2020, 87, 105986; Chem. Mater. 2021, 33, 12, 4541–4550]
Reviewer 2 Report
The manuscript by Qu et al. studies here Physical Aging Behavior of the Side Chain of a Conjugated Polymer PBTTT. The results are interesting and well analyzed. The methods for investigation are appropriate and state-of-the-art. Therefore, I recommend publication with minor changes.
Examples,
Figure 2 : 183K
Figure 3: Arrhnius
use of hyphenated words were inappropriately done in abstract and throughout the manuscript.
